# Effects of Fat Type and Exogenous Bile Acids on Growth Performance, Nutrient Digestibility, Lipid Metabolism and Breast Muscle Fatty Acid Composition in Broiler Chickens

**DOI:** 10.3390/ani12101258

**Published:** 2022-05-13

**Authors:** Shixia Geng, Yuxin Zhang, Aizhi Cao, Ying Liu, Yuting Di, Juntao Li, Qianqian Lou, Liying Zhang

**Affiliations:** State Key Laboratory of Animal Nutrition, Ministry of Agriculture and Rural Affairs Feed Industry Centre, China Agricultural University, Beijing 100193, China; gengshixia@163.com (S.G.); s20203040632@cau.edu.cn (Y.Z.); aizhicao78@163.com (A.C.); liuyingcau2021@163.com (Y.L.); dyt971004@163.com (Y.D.); wacau@163.com (J.L.); shubaobaodeweilai@126.com (Q.L.)

**Keywords:** bile acid, broiler chicken, fat type, lipid metabolism, performance

## Abstract

**Simple Summary:**

Exogenous BAs can serve as emulsifiers and when supplemented in broiler chicken diets, they can facilitate the absorption of fat and fat-soluble nutrients. As far as we know, no research compared the impacts of BAs on broiler chickens fed with animal fat and vegetable oil. Therefore, the current research aimed to explore the impacts of fat type and BAs on growth performance, nutrient digestibility, lipid metabolism, and breast muscle fatty acids composition in broiler chickens. The results showed that the positive impacts of BAs were obvious for broiler chickens fed diets with lard compared with soybean oil.

**Abstract:**

The current study aimed to explore the effects of fat type and exogenous bile acids (BAs) on growth performance, nutrient digestibility, lipid metabolism, and breast muscle fatty acids composition in broiler chickens. A total of 432 one-day-old Arbor Acres male broilers were stochastically distributed to a 2 × 2 factorial design comprised of two fat types (soybean oil and lard) and two levels of BAs (0 and 80 mg/kg) included in diets, totaling 4 treatments of 6 replicate pens with 18 chicks per replicate pen. Compared with treatments with soybean oil, dietary inclusion of lard increased the digestibility of ether extract (EE) in diets and the percentage of breast muscle on d 42, and increased the level of serum triglycerides and decreased serum alanine aminotransferase (ALT) activity on d 21 (*p* < 0.05). The level of saturated fatty acids, monounsaturated fatty acids (MUFAs), and the n-6 to n-3 polyunsaturated fatty acids ratio in breast muscle were also increased (*p* < 0.05) when feeding lard versus soybean oil. Dietary supplementation with BAs elevated average daily gain and reduced the ratio of feed to gain at d 0–21 and 0–42, significantly (*p* < 0.05). The digestibility of EE in diets and the percentage of breast muscle on d 42 were also increased by BAs (*p* < 0.05). Serum total cholesterol content as well as the percentage of abdominal fat on d 42, and ALT activity on d 21 were decreased when BAs were fed (*p* < 0.05). The concentration of total fatty acids, saturated fatty acids, and MUFAs of breast muscle were decreased by BAs. These results indicate that BAs can increase growth performance and nutrient digestibility, elevate carcass characteristics, and improve lipid metabolism, and their effects on nutrient digestibility and carcass characteristics were more pronounced in broiler chickens fed diets with lard.

## 1. Introduction

The energy value of fats is at least twice those of protein and carbohydrates [1,2]. They are preferentially included in the feeds of broiler chickens to satisfy the high energy requirements of growth. However, their inclusion with fats and oils can lead to excessive accumulation of fat in the carcass of broiler chickens, especially in abdominal and visceral areas, which are one of the undesirable consequences of selection for increased growth of broiler chickens [3]. Obese broiler chickens reduce the purchase intention of consumers who are concerned about nutritional and healthy aspects of meat, and decrease the net return for producers as abdominal fat is non-profitable. Under these circumstances, producers of modern broiler chickens have taken many strategies, including an increase in the proportion of unsaturated fatty acids (UFAs) in dietary fat [4,5], and the supplementation of exogenous enzymes [6,7] and emulsifiers [8,9]. The potential of bile acids (BAs) as emulsifiers has been investigated previously by others [10,11].

In general, BAs are a series of compounds composed of a hydrophobic face, a concave hydrophilic face, and a lateral chain [12]. They are constituted by primary BAs such as cholic acid (CA) and chenodeoxycholic acid (CDCA), as well as secondary BAs such as deoxycholic (DCA) and lithocholic acid (LCA), with diverse numbers and positions of hydroxyl groups [13,14]. Due to this amphipathic nature, BAs can work as emulsifiers to promote the utilization of fat and fat-soluble vitamins [15]. Studies have indicated that supplementary BAs in animal diets can elevate nutrient digestibility [11,16], growth performance [10,16], and decrease abdominal fat deposition [17,18].

The dietary sources of fats and oils mainly include vegetable oils and animal fats that are individually abundant with UFAs and saturated fatty acids (SFAs) [19]. It has been reported that more fat accumulates in the body of animals fed diets that include animal fats compared to those with vegetable oils [4,19], and the absorption of animal fats was lower than that of vegetable oils [20]. We have previously demonstrated that dietary supplementation with 80 mg/kg porcine BAs decreased the abdominal fat deposition of broiler chickens when feeding diets including lard [17]. As far as we know, the effects of BAs on broiler chickens fed diets including different fats with different degrees of saturation have not been reported before. Therefore, this study chose soybean oil and lard as sources of UFAs and SFAs to explore the impacts of dietary exogenous BAs on performance, nutrient digestibility, lipid metabolism, and fatty acids profile of breast muscle in broiler chickens fed diets included with fats with different degrees of saturation.

## 2. Materials and Methods

The current study was performed in the FengNing Research Unit of China Agricultural University (Academician Workstation in Chengdejiuyun Agricultural and Livestock Co., Ltd., Chengde, Hebei, China) and all animal process was authorized by the Institutional Animal Care and Use Committee of China Agricultural University (approval code: AW62102202-1-1; 19 January 2022). 

### 2.1. Preparation for BAs

Shandong Longchang Animal Health Care Co. Ltd. (Dezhou, China) provided us with the experimental porcine BAs; the total BA level was over 95.0%. Their composition was tested by high-performance liquid chromatography: HDCA (70.87%), CDCA (19.95%), and HCA (7.53%). These porcine BAs were processed from porcine bile through a series of physicochemical means.

### 2.2. Experimental Broiler Chickens, Design, Management, and Diets

A total of 432 (average body weight, 39.0 ± 0.54 g) one-day-old male Arbor Acres broilers were bought from Arbor Acres Poultry Breeding Company (Beijing, China). They were stochastically distributed into a 2 × 2 factorial design composed of 2 fats (soybean oil and lard) and 2 BAs levels (0 and 80 mg/kg) included in diet, totaling 4 treatments with 6 replicate pens of 18 broiler chickens. Experimental treatments were listed as follows: SO (diets included with soybean oil), LO (diets included with lard), SB (diets included with soybean oil supplemented with 80 mg/kg BAs), and LB (diets included with lard supplemented with 80 mg/kg BAs).

Broiler chickens received water and feed ab libitum, and lights were turned off for an hour every day. During the first three days, room temperature started at 35 °C and was lowered 3 °C every week to finally achieve 24 °C. The relative humidity was 45–55%. At the end of the first and second week, broiler chickens were vaccinated against Newcastle disease. Inactivated infectious bursa disease vaccine was inoculated to all chickens when the second and fourth weeks finished.

Nutrients formulated in experimental diets satisfied even surpassed the suggested values suggested by the NRC [21] (Table 1). Association of Official Analytical Chemists [22] were used when testing nutrients in diets including total phosphorus (using method 995.11), calcium (using method 927.02), ether extract (EE) (using method 920.39), dry matter (DM) (using method 934.01), crude protein (CP) (using method 988.05), and amino acids (using method 994.12) (Table 2). Before acid hydrolysis by 6 M HCl, performic acid oxidation was administrated to determine dietary methionine determination.

### 2.3. Growth Performance, Nutrient Digestibility, and Sample Collection

Weight of body and weight of feed intake were noted on d 21 and d 42. Average daily gain (ADG), average daily feed intake (ADFI), and the ratio of feed to gain (F:G) at d 0–21, d 21–42, and d 0–42 were computed with the above corresponding values, following the formulas: ADG = (Average finish weight—average start weight)/Age (days); ADFI = Total feed consumption per chick/Age (days); F:G = ADFI/ADG.

Diets were added with chromic oxide during d 35 to d 42 at a level of 3.0 g/kg feed, serving as the marker to measure the digestibility of DM, CP, EE, and gross energy (GE). About 100 g of feces from every replicate was collected on d 40, 41, and 42. The feces were pooled within a replicate and dried at 65 °C, then DM, CP, EE, and GE were analyzed according to the methods that are the same as that used for the analysis of diets. The GE and chromium in diets and feces were measured with 6400-Automatic Isoperibol Calorimeter (PARR, Moline, IL, USA) and atomic absorption spectrophotometer (Z-5000, Hitachi, Tokyo, Japan), respectively. Calculation formula of digestibility: digestibility (%) = 100% − 100% × (Cr _feed_ × Nutrient _feces_)/(Cr _feces_ × Nutrient _feed_).

On the 21st and 42nd day, 6 broiler chickens from each replicate with weight approximating the average weight were picked to collect serum samples from the wing vein after 12 h fast. The anticoagulant-free vacutainer tubes were used to collect the venous blood samples and the samples were allowed 30 min conservation at room temperature. Afterward, they were centrifugated for 15 min with speed of 3600× *g*. The serum was sucked out and stored under −20 °C to further test.

Broiler chickens picked from each replicate were stunned using 60% concentration of CO_2_ and then they were euthanized via jugular vein bleeding. About 20 g muscle sample from the left breast of broiler chickens was taken and reserved under −20 °C. On the 42nd day, live weight, carcass weight, eviscerated weight, and weight of thigh muscle, breast, and abdominal fat were measured. The live weight was used for computing the dressing percentage and the eviscerated percentage, and the eviscerated weight was used to compute the breast, leg muscle, and abdominal fat percentage.

### 2.4. Metabolites in Serum

Biochemical parameters in serum were composed of the content of total cholesterol (TC), triglycerides (TG), low-density lipoprotein cholesterol (LDL-C), and high-density lipoprotein cholesterol (HDL-C), and the activity of aminotransferase, i.e., alanine aminotransferase (ALT) and aspartate aminotransferase (AST). These parameters were tested by commercial kits (Biosino Biotechnology and Science Inc., Beijing, China) with an Automatic Biochemical Analyzer (Hitachi 7160, Hitachi High-Tech Corporation, Tokyo, Japan).

### 2.5. Fatty Acids Composition of Breast Muscle

About 20 g of breast muscle from each repeat collected on day 42 were lyophilized in a lyophilizer for 60 h and the lyophilized muscle samples were smashed in the pulverizer for fatty acid analysis. Fatty acid profiles of breast muscle were analyzed with gas chromatography (Agilent Technologies 6890 series, Wilmington, DE, USA) and followed instructions reported by Sukhija and Palmquist [23]. The pre-processing process is as follows: First, about 0.2 g sample was added into the hydrolysis tube, then 4 mL 10% chloroacetylmethanol solution, 1 mL 1.0 mg/mL undecanoic acid (as internal standard), and 1 mL n-hexane were added to the tube, which was subsequently put on the cap and placed in an 80 °C-thermostat water bath for 2 h. After cooling at room temperature, 5 mL of 7% potassium carbonate was added, shaken homogeneously. Then they were centrifuged at 1000 r/min for 5 min. The clear supernatant was filter across a 0.2 µm membrane and then injected into the vial to be detected by gas chromatography. Procedural ramp-up was as follows: 50 °C, 20 min; from 50 to 170 °C with rate of 10 °C/min for 10 min; from 170 to 180 °C with rate of 4 °C/min for 10 min; and then from 180 to 220 °C with speed of 2 °C/min for 10 min. The temperature of the detector was set at 270 °C; hydrogen 40 mL/min; airflow 400 mL/min; and helium 40 mL/min. Each took 50 min.

### 2.6. Statistical Analysis

Data of this study were analyzed as a 2 × 2 (fat type × BAs) factorial design of treatments by two-way ANOVA with the GLM procedure of SAS 9.2 (SAS Institute Inc., Cary, NC, USA). This model was comprised of the impacts of fat type and BAs as well as the interaction of fat type and BAs. If the interaction between fat type and BAs tested differently, significantly (*p* < 0.05), then a comparison of the means was conducted using Tukey’s multiple range test. Statistical significance was based on a *p*-value less than 0.05.

## 3. Results

### 3.1. Growth Performance

The type of fat did not influence growth performance (*p* > 0.05). The supplementation with BAs elevated ADG and reduced F:G at d 22–42 and d 0–42 (*p* < 0.05). The interaction between fat type and BAs was not observed (*p* > 0.05, Table 3).

### 3.2. Carcass Characteristics

The breast muscle percentage of broiler chickens was increased by diets with lard in comparison with diets with soybean oil (*p* < 0.05, Table 4). Dietary BAs elevated the ratio of breast muscle and reduced the ratio of abdominal fat, respectively (*p* < 0.05).

### 3.3. Nutrient Digestibility

Diets included with lard increased EE digestibility in broiler chickens compared to diets with soybean oil (*p* < 0.05, Table 5). BAs elevated the EE and GE digestibility (*p* < 0.05). An interaction between fats and BAs was not found in nutrient digestibility (*p* > 0.05).

### 3.4. Biochemical Parameters in Serum

Compared with diets included with lard, diets with soybean oil decreased serum TG content and increased the activity of ALT in broiler chickens on d 21 (*p* < 0.05, Table 6). At the same time, BAs reduced serum ALT activity on d 21. The interaction between fat type and BAs was observed in serum ALT activity on d 21, which was significantly increased in the SO group compared with the other groups (*p* < 0.05). On d 42, BAs lowered the content of serum TC and HDL-C (*p* < 0.05).

### 3.5. Breast Muscle Fatty Acids Profile

Compared with diets with lard, soybean oil diets elevated the concentration of polyunsaturated fatty acids (PUFAs), n-6 PUFAs, n-3 PUFAs, and P/S, and decreased the n-6 to n-3 PUFAs ratio of breast (*p* < 0.05, Table 7). BAs decreased the content of monounsaturated fatty acids (MUFAs), SFAs, and total fatty acids of breast muscle and the interaction between fats and BAs was noted (*p* < 0.05). The level of MUFAs, SFAs, and total fatty acids were comparable in soybean oil groups with or without BAs, whereas BAs decreased their level when feeding broilers with diets that included lard (*p* < 0.05).

## 4. Discussion

BAs are synthesized through an enzyme-catalyzed reaction from cholesterol in liver perivenous hepatocytes in vivo [13]. Upon food stimulation, BAs are secreted into the duodenum and play a role of vital importance in the digestion and utilization of lipids [24]. Nowadays, in order to satisfy the need of contemporary fast-growing birds, fats are more often used in feed; however, they cannot be well digested and absorbed by broiler chickens, especially the newly hatched ones, due to limited BAs or lipase secretion [2]. Thus, exogenous BAs are often supplemented into broiler diets as emulsifiers to improve the performance of broiler chickens. Parsaie et al. [10] found that the addition of 500 mg/kg cholic acid in the broiler diet elevated their feed intake as well as weight gain. Alzawqari et al. [11] supplemented 0.5% ox bile in wheat-based diets of broiler chickens and observed that ox bile elevated weight gain and feed efficiency. In this study, dietary supplemental 80 mg/kg porcine BAs elevated ADG as well as decreased F:G. As an essential emulsifier, exogenous BAs not only increased the digestibility of EE in broiler chickens in this study, but also increased the utilization rate of gross energy in diets. Maisonnier et al. [16] and Shoaib et al. [7] supplemented with taurocholic acid and a mixture of hyocholic acid, hyodeoxycholic acid, and CDCA in the broiler diet, respectively, and observed that fat digestibility was increased in these treatment groups. Jansen et al. [25] found that the addition of soybean and rapeseed lysolecithin as an emulsifier increased the metabolic energy value of diets. Thus, dietary supplemental BAs promote growth performance in broiler chickens partly through improving the utilization of EE and GE in the diets of broiler chickens.

Besides the positive effect of BAs on lipids digestion, the degree of unsaturation of dietary fats also affects the intestinal digestion of lipids. Vegetable oil containing high concentrations of UFAs is easily digested and better absorbed than animal fat [26]. However, in the current study, EE digestibility of broiler chickens fed with lard was higher compared to that with soybean oil, and BAs increased EE digestibility to a greater extent in lard groups compared with soybean oil groups (2.16 vs. 1.24). These indicated that the effects of BAs as an emulsifier might be more pronounced for animal fats. Similarly, Garrett and Young et al. [27] discovered that the solubilization and absorption of SFAs are more negatively impacted by a lack of bile salts, than those of UFAs. Jansen et al. [25] found that the supplementation with soybean and rapeseed lysolecithins as emulsifiers facilitated in vitro hydrolysis of lard while they had no effect on the hydrolysis of soybean oil. They also found that the addition of soybean and rapeseed lysolecithin elevated the metabolic energy value more in groups with lard compared to that in groups with soybean oil, but no significant difference existed in the metabolic energy value between the soybean oil group and the lard group after supplementing soybean and rapeseed lysolecithin. In this study, diets included with soybean oil or lard did not affect the growth of broiler chickens. Similar observations were made by Ferrer et al. [28] using linseed oil, sunflower oil, and lard, and by Fébel et al. [29], using lard, and soybean, sunflower, and linseed oil in Ross chickens.

The carcass characteristic is one of the essential indexes for the economic efficiency of broiler chickens. Previous studies reported that exogenous BAs could improve carcass characteristics through lowering valueless abdominal fat accumulation and elevating muscle production such as leg and breast [17,18]. This study also showed that BAs could lower the ratio of abdominal fat and increase the ratio of breast muscle. Similar to EE digestibility, BAs elevated the breast muscle percentage more in broiler chickens fed lard diets than that in broiler chickens fed soybean oil diets (1.77 vs. 0.77). BAs elevated body weight gain as well as the breast muscle percentage but decreased abdominal fat deposition, and this implied that exogenous BAs might be more beneficial for muscle growth. Sasaki et al. [30] found that activation of the receptor of BAs in skeletal muscles promoted muscle hypertrophy and differentiation in mice, which revealed the possible mechanism of effects of BAs on muscle at the molecular level. Compared with vegetable oils, animal fats are often thought to be more easily deposited in the body. Crespo and Esteve-Garcia [4] administered different types of fats including animal fats and vegetable oils in a broiler diet at levels of 6% or 10% during the grower phase, and observed that the percentage of abdominal fat of those eating sunflower and linseed oils was reduced compared with broiler chickens eating tallow diets. However, the abdominal fat rate was not influenced by fat type with the inclusion level of 3% during the grower phase in this study. Zollitsch et al. [26] also noted that the inclusion of 3.5% fats including soybean oil, rapeseed oil, and a blend of animal and vegetable fat did not influence the rate of abdominal fat in chickens. We believe that the discrepancy of fat deposited in broiler chickens fed with animal fats and vegetable oils does not appear until the inclusion level of fat has been increased to a certain extent. Further experiments need to be conducted with different degrees of saturation and levels of fat under the same condition to confirm these results.

De novo fatty acids synthesis mainly occurs in the liver of broiler chickens, and triglycerides are transported from the liver to peripheral tissues through LDL and from peripheral tissues to the liver through HDL [31,32]. In this study, dietary inclusion with soybean oil decreased serum TG level on d 21 compared with lard. Ortiz-Munoz et al. [33] and Liu et al. [34] discovered that diets that included fat enriched with UFA showed lower serum TGs as well. Meanwhile, supplementation with BAs decreased the level of serum TC and HDL on d 42 in this study. Ge et al. [18] observed that serum TG content was decreased by BAs while the level of serum HDL-C was not changed, which might be due to using diets with a high content of energy in their study. The decline of HDL in the current study occurred with the overall decrease in lipid content in serum and the abdominal area, and subsequently, fat transport via HDL or LDL was restricted. From the above, supplementation with BAs decreased fat deposition in the body of broiler chickens and this might be achieved by regulating lipid metabolism. BAs have been proved to participate in the expression of hepatic lipogenesis by nuclear receptor-farnesoid X receptor (FXR) [35,36,37]. CDCA has been reported as an effective ligand combining FXR [38] and researchers have found that dietary supplemental CDCA or porcine BAs decreased the expression level of hepatic lipogenesis genes in broiler chickens [39,40]. CDCA is also the main component of the porcine BAs used in this study. Therefore, porcine BAs might reduce fat deposition and improve the lipid homeostasis of broiler chickens by inhibiting the expression of genes connected with hepatic fatty acids synthesis.

BAs, acting as signal molecules, can improve not only lipid metabolism but also glucose utilization and inflammation through FXR and G-protein-coupled bile acid receptors. Furthermore, bile acid receptors have become a popular potential treatment for liver diseases such as nonalcoholic fatty liver disease, liver inflammation, and fibrosis, and are beneficial for liver health [41]. Liver-related enzymes such as alanine aminotransferase (ALT) and aspartate aminotransferase (AST) are essential parameters for the health of the liver. They commonly stay in cells of the liver, but they are launched into serum when suffering from cell damage [42]. In this study, BAs decreased the activity of ALT, which indicated that dietary supplemental BAs were beneficial for the liver health of broiler chickens, and this positive effect might be related to BA receptors. Meanwhile, serum ALT activity of broiler chickens fed diets with soybean oil was higher than that of broiler chickens fed with lard. Fébel et al. [29] observed that diets that included soybean oil elevated serum AST activity compared with diets that included lard. Both Fébel et al. [29] and Cherian et al. [43] explored the effect of the degree of unsaturated fats included in diets on the antioxidant response of broiler chickens and discovered that the level of the product of lipid peroxidation-malondialdehyde (MDA) increased in the liver in those fed diets with fats with a higher degree of unsaturation. Aldehyde-derived protein modification has been widely verified to be a key factor in liver injury [44,45]. Therefore, the negative effect on the liver caused by an elevated level of cytotoxic MDA in response to lipid peroxidation partly explains why serum ALT activity was higher in broiler chickens fed with soybean oil than that with lard in this study.

The nutritional value and characteristic flavors of meat are related to their fatty acid content. It has been reported that UFAs can serve as essential precursors to produce fatty aromas and characteristic flavors of chicken meat through oxidative decomposition and the Maillard reaction [46]. Additionally, UFAs can lower the risk of suffering from cardiovascular diseases in humans [47]. Recent literature has also shown that the consumption of food with a higher n-6 to n-3 PUFAs percentage tends to lead to the development of diseases such as cardiovascular disease and autoimmune diseases, because n-3 fatty acids were found to decrease fat deposition [48] and n-6 fatty acids were found to increase the cellular triglyceride content [49]. In this study, dietary inclusion with soybean oil reduced the n-6 to n-3 PUFA ratio and elevated the PUFAs to SFAs ratio in the breast of broiler chickens, in comparison with diets included with lard. These variations were consistent with the results of studies by other authors [26,29]. Notably, supplementation of BAs lowered the level of total fatty acids, SFAs and MUFAs, and interaction effects existed on these parameters with the minimum at group LB. Moreover, BAs decreased the level of some n-6 fatty acids including dohomo-γ-linolenic acid (C20:3n6) and arachidonic acid (C20:4n6). These changes should be beneficial for the nutritive value of breast muscle. To the best of our knowledge, no literature explored the impacts of BAs on the profile of fatty acids of the breast muscle of broiler chickens. Nie et al. [50] found that specific bile acids could restrict the uptake of fatty acids by primary hepatocytes through the fatty acid transport protein 5. Unlike mammalian species whose adipose tissue is essential to synthesize fatty acids, the liver is the most important place for avian species to synthesize fatty acids and their synthesis is influenced by the intake of dietary fatty acids [51]. Moreover, for poultry, the absorption of dietary fats with certain fatty acid profiles usually reflects the fatty acid pattern deposited in tissues [52]. Thus, the absorption, de novo synthesis, and oxidation of fatty acids determined the pattern of fatty acids in meat to a certain extent [53]. The reduction in fatty acids transported from diets to the liver via the portal vein might cause a decreased level of fatty acids transported from the liver to the breast muscle, and this could be the possible mechanism.

## 5. Conclusions

Dietary supplementation with BAs increased growth performance and nutrient digestibility, improved carcass characteristics, as well as reduced lipid deposition. Moreover, the effects of BAs were more obvious for broiler chickens fed diets with lard than for those with soybean oil. Thus, BAs work better with dietary fats with a higher degree of saturation.

## Figures and Tables

**Table 1 animals-12-01258-t001:** The levels of ingredients of basal diets (%).

Ingredients	d 0–d 21	d 22–d 42
Corn	60.13	61.53
Soybean meal	32.50	31.70
Fish meal	2.00	0.00
Soybean oil/Lard	1.50	3.00
Dicalcium phosphate	1.50	1.70
Limestone	1.34	1.15
DL-methionine (98%)	0.23	0.12
Sodium chloride	0.30	0.30
Vitamin–mineral Premix ^1^	0.50	0.50

^1^ Provided as the content in one kilogram feed: axerophthol, 9000 IU; cholecalciferol, 3000 IU; tocopherol, 24 mg; vitamin K_3_, 1.8 mg; vitamin B_1_, 2.0 mg; vitamin B_2_, 5.0 mg; vitaminB_6_, 3.0 mg; vitamin B_12_, 0.1 mg; nicotinic acid, 40 mg; biotin, 0.05 mg; pantothenic acid, 15 mg; choline chloride, 500 mg; folic acid, 1.0 mg; iron, 80 mg; copper, 20 mg; selenium, 0.35 mg; zinc, 90 mg; iodine, 0.35 mg; manganese, 80 mg.

**Table 2 animals-12-01258-t002:** Analyzed levels of nutrients of diets (%) ^1^.

Nutrients Levels (%) ^2,3^	SO	SB	LO	LB
Starter phase (0–21 d)				
ME (kcal/kg)	2960	2960	2971	2971
Crude protein	21.06	21.17	21.57	21.48
Calcium	0.86	0.94	0.88	0.90
Total phosphorus	0.61	0.63	0.61	0.62
Methionine + cystine	0.82	0.85	0.83	0.82
Lysine	1.20	1.16	1.20	1.21
Grower phase (22–42 d)				
ME (kcal/kg)	3020	3020	3042	3042
Crude protein	20.31	19.74	20.13	19.87
Calcium	0.82	0.81	0.77	0.79
Total phosphorus	0.56	0.57	0.56	0.56
Methionine + cystine	0.75	0.77	0.76	0.74
Lysine	1.14	1.11	1.13	1.12

^1^ The level of nutrients was detected values with 2 determinations except for metabolizable energy. ^2^ SO = diets included with soybean oil; LO = diets included with lard; SB = diets included with soybean oil supplemented with bile acids; LB = diets included with lard supplemented with bile acids. ^3^ ME = metabolizable energy, the value of ME was obtained by calculation.

**Table 3 animals-12-01258-t003:** Effects of bile acids supplemented in diet included with different types of fat on growth performance in broiler chickens.

Items ^1,2^	SO	SB	LO	LB	SEM ^3^	*p*-Value		
						Fat Type	Bile Acids	Fat Type × Bile Acids
Bodyweight								
0 d (g)	38.71	38.65	39.12	38.71	0.54	0.67	0.67	0.76
21 d (g)	672.39	659.59	658.75	656.67	10.66	0.45	0.50	0.62
42 d (kg)	2.13	2.22	2.13	2.17	0.02	0.22	0.01	0.37
Day 0–21								
ADG (g/d)	30.20	29.57	29.51	29.43	0.50	0.42	0.49	0.59
ADFI (g/d)	40.42	39.87	39.36	38.98	0.67	0.17	0.49	0.90
F:G	1.34	1.35	1.33	1.32	0.01	0.33	0.96	0.50
Day 22–42								
ADG (g/d)	69.54	74.15	69.82	71.98	1.08	0.40	<0.01	0.28
ADFI (g/d)	113.74	116.84	111.45	113.60	1.48	0.09	0.10	0.75
F:G	1.64	1.58	1.60	1.58	0.01	0.13	<0.01	0.13
Day 0–42								
ADG (g/d)	48.91	50.84	48.68	49.67	0.51	0.19	0.04	0.37
ADFI (g/d)	75.52	76.68	73.84	74.67	0.82	0.05	0.25	0.85
F:G	1.54	1.50	1.52	1.50	0.01	0.06	<0.01	0.16

^1^ ADG = average daily gain; ADFI = average daily feed intake; F:G = ratio of feed to gain. ^2^ SO = diets included with soybean oil; LO = diets included with lard; SB = diets included with soybean oil supplemented with bile acids; LB = diets included with lard supplemented with bile acids. ^3^ SEM = total standard error of means (*n* = 6).

**Table 4 animals-12-01258-t004:** Effects of bile acids supplemented in diet included with different types of fat on carcass characteristics in broiler chickens (%).

Items ^1^	SO	SB	LO	LB	SEM ^2^	*p*-Value		
						Fat Type	Bile Acids	Fat Type × Bile Acids
Dressing percentage	93.20	92.63	92.90	92.55	0.60	0.76	0.46	0.86
Eviscerated percentage	72.99	73.00	73.21	74.22	0.65	0.27	0.44	0.45
Breast muscle percentage	23.29	24.09	24.09	25.86	0.73	0.04	0.03	0.68
Leg muscle percentage	23.00	22.94	22.94	22.60	0.65	0.22	0.38	0.43
Abdominal fat percentage	1.30	1.12	1.21	0.90	0.10	0.11	0.02	0.48

^1^ SO = diets included with soybean oil; LO = diets included with lard; SB = diets included with soybean oil supplemented with bile acids; LB = diets included with lard supplemented with bile acids. ^2^ SEM = total standard error of means (*n* = 6).

**Table 5 animals-12-01258-t005:** Effects of bile acids supplemented in diet included with different types of fat on dietary nutrient digestibility in broiler chickens (%).

Items ^1,2^	SO	SB	LO	LB	SEM ^3^	*p*-Value		
						Fat Type	Bile Acids	Fat Type × Bile Acids
EE	82.10	83.34	82.76	84.92	0.51	<0.05	<0.01	0.38
CP	57.44	60.49	61.38	62.00	1.30	0.06	0.18	0.37
DM	71.13	71.76	71.32	72.05	0.50	0.64	0.20	0.91
GE	76.04	76.84	75.32	76.42	0.41	0.19	0.04	0.73

^1^ EE = ether extract; CP = crude protein; DM = dry matter; GE = gross energy. ^2^ SO = diets included with soybean oil; LO = diets included with lard; SB = diets included with soybean oil supplemented with bile acids; LB = diets included with lard supplemented with bile acids. ^3^ SEM = total standard error of means (*n* = 6).

**Table 6 animals-12-01258-t006:** Effects of bile acids supplemented in diet included with different types of fat on serum biochemical parameters in broiler chickens.

Items ^1,2^	SO	SB	LO	LB	SEM ^3^	*p*-Value		
						Fat Type	Bile Acids	Fat Type × Bile Acids
Day 21								
TC, mmol/L	3.48	3.06	3.44	3.32	0.18	0.53	0.14	0.42
TG, mmol/L	0.52	0.50	0.60	0.67	0.04	<0.01	0.53	0.37
HDL-C, mmol/L	2.35	2.25	2.39	2.32	0.10	0.33	0.57	0.85
LDL-C, mmol/L	0.76	0.68	0.77	0.78	0.07	0.42	0.59	0.54
GLU, mmol/L	10.03	9.69	10.02	9.39	0.24	0.54	0.06	0.56
ALT, U/L	4.63 ^a^	2.37 ^b^	3.05 ^b^	2.58 ^b^	0.31	0.04	<0.01	<0.01
AST, U/L	251.07	230.32	236.38	244.75	9.55	0.99	0.52	0.14
Day 42								
TC, mmol/L	3.47	3.06	3.47	3.12	0.11	0.25	0.02	0.27
TG, mmol/L	0.39	0.42	0.45	0.41	0.03	0.39	1.00	0.18
HDL-C, mmol/L	2.57	2.21	2.49	2.40	0.08	0.48	<0.01	0.10
LDL-C, mmol/L	0.76	0.66	0.77	0.75	0.04	0.26	0.18	0.36
GLU, mmol/L	10.83	10.92	9.73	10.63	0.53	0.21	0.36	0.46
ALT, U/L	4.23	2.90	3.15	3.40	0.48	0.55	0.27	0.12
AST, U/L	354.03	351.82	358.23	368.30	34.72	0.77	0.91	0.86

^a,b^ Means that statistical significant difference with different superscripts in a row existed. ^1^ TC = total cholesterol; TG = total triglyceride; HDL-C = high-density lipoprotein cholesterol; LDL-C = low-density lipoprotein cholesterol; GLU = glucose; ALT = alanine aminotransferase; AST = aspartate aminotransferase. ^2^ SO = diets included with soybean oil; LO = diets included with lard; SB = diets included with soybean oil supplemented with bile acids; LB = diets included with lard supplemented with bile acids. ^3^ SEM = total standard error of means (*n* = 6).

**Table 7 animals-12-01258-t007:** Effects of bile acids supplemented in diet included with different types of fat on breast muscle fatty acids composition in broiler chickens (mg/g).

Items ^5^	SO	SB	LO	LB	SEM ^6^	*p*-Value		
						Fat Type	Bile Acids	Fat Type × Bile Acids
C14:0	0.15 ^b^	0.14 ^b^	0.26 ^a^	0.19 ^b^	0.02	0.01	<0.01	<0.01
C14:1	0.03	0.03	0.04	0.02	0.01	0.62	0.15	0.08
C16:0	7.83 ^ab^	7.89 ^ab^	8.85 ^a^	4.94 ^b^	0.79	0.24	0.03	0.02
C16:1	0.88 ^ab^	0.93 ^ab^	1.31 ^a^	0.75 ^b^	0.13	0.37	0.07	0.03
C17:0	0.06 ^a^	0.05 ^a^	0.06 ^a^	0.03 ^b^	0.01	0.01	<0.01	0.05
C18:0	3.70 ^a^	3.51 ^a^	3.73 ^a^	2.65 ^b^	0.30	0.04	0.01	0.03
C18:1n9c	10.16 ^ab^	10.10 ^ab^	13.03 ^a^	6.92 ^b^	1.15	0.89	0.01	0.02
C18:2n6c	9.79	9.65	6.28	3.29	0.92	<0.01	0.11	0.14
C18:3n3	0.75	0.76	0.25	0.11	0.08	<0.01	0.42	0.38
C20:0	0.06 ^a^	0.05 ^a^	0.07 ^a^	0.03 ^b^	0.01	0.09	<0.01	0.03
C20:1	0.13 ^b^	0.12 ^b^	0.19 ^a^	0.11 ^b^	0.01	0.06	<0.01	<0.01
C21:0	0.15 ^a^	0.15 ^a^	0.15 ^a^	0.09 ^b^	0.01	0.04	0.02	0.03
C20:3n6	0.33 ^ab^	0.35 ^a^	0.34 ^ab^	0.23 ^b^	0.03	0.07	0.11	0.04
C20:4n6	1.69 ^a^	1.57 ^a^	1.57 ^a^	0.92 ^b^	0.12	<0.01	<0.01	0.04
C20:3n3	0.05	0.04	0.03	0.02	0.01	<0.01	<0.01	0.82
C22:0	0.05	0.05	0.05	0.04	0.01	0.20	0.20	0.20
C20:5n3	0.07	0.07	0.06	0.05	0.01	0.03	0.29	0.29
C24:1	0.07 ^b^	0.07 ^b^	0.09 ^a^	0.06 ^b^	0.01	0.25	<0.01	<0.01
C22:6n3	0.18	0.15	0.16	0.10	0.02	0.03	<0.01	0.43
Others	0.19	0.16	0.21	0.14	0.01	1.00	<0.01	0.08
total fatty acids	36.31 ^a^	35.85 ^a^	36.70 ^a^	20.10 ^b^	3.47	0.04	0.02	0.03
SFA ^1^	12.12 ^a^	11.97 ^a^	13.32 ^a^	7.48 ^b^	1.13	0.16	0.02	0.02
MUFA ^2^	11.31 ^ab^	11.29 ^ab^	14.69 ^a^	7.89 ^b^	1.29	0.99	0.01	0.02
PUFA ^3^	12.88	12.59	8.70	4.72	1.14	<0.01	0.08	0.12
n-6 PUFA	11.81	11.57	8.18	4.44	1.05	<0.01	0.07	0.11
n-3 PUFA	1.06	1.01	0.49	0.27	0.09	<0.01	0.15	0.35
n-6/n-3	11.10	11.61	16.75	16.40	0.56	<0.01	0.89	0.45
P/S ^4^	1.07	1.05	0.66	0.64	0.02	<0.01	0.40	0.89

^a,b^ Means that statistical significant difference with different superscripts in a row existed. C14:0 = myristic acid; C14:1 = myristoleic acid; C16:0 = palmitic acid; C16:1 = palmitoleic acid; C17:0 = margaric acid; C18:0 = stearic acid; C18:1n9c = oleic acid; C18:2n6c = linoleic acid; C18:3n3 = linolenic acid; C20:0 = eicosanoic acid; C20:1 = eicosenoic acid; C21:0 = henicosanoic acid; C20:3n6 = dihomogamma linolenic acid; C20:4n6 = arachidonic acid; C20:3n3 = eicosatrienoic acid; C22:0 = behenic acid; C20:5n3 = eicosapentaenoic acid; C24:1 = nervonic acid; C22:6n3 = docosahexaenoic acid; ^1^ SFA = saturated fatty acid; ^2^ MUFA = monounsaturated fatty acid; ^3^ PUFA = polyunsaturated fatty acid; ^4^ P/S = ratio of polyunsaturated fatty acid to saturated fatty acid. ^5^ SO = diets included with soybean oil; LO = diets included with lard; SB = diets included with soybean oil supplemented with bile acids; LB = diets included with lard supplemented with bile acids. ^6^ SEM = total standard error of means (*n* = 6).

## Data Availability

The data presented in this study are available on request from the corresponding author.

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
