# Peer review of "Effects of Fat Type and Exogenous Bile Acids on Growth Performance, Nutrient Digestibility, Lipid Metabolism and Breast Muscle Fatty Acid Composition in Broiler Chickens"

_animals, 2022, doi:10.3390/ani12101258_

Round 1

Reviewer 1 Report

The manuscript seems interesting. However, additional data for health condition is necessary. For example, blood biochemical parameters related to health, not only lipid profiles presented here, but also glucose, ALT, AST, etc. which are general indicators in medical fields of humans, are necessary. In general, lard is not good for our health. 

Reviewer 2 Report

Dear Authors, I suggest some changes to the manuscript:

  1. The subject contains a description of "broilers", however they can be: broiler chickens, broiler ducks, broiler geese ... The topic in its current form is not precise. Then you can read in the abstract that these are Arbor Acres broilers ... Not everyone can know this line. Please specify the bird species exactly.
  2. in the Introduction section, also use the species names, not just "broiler".
  3. in the materials before section 2.1. Please describe the licensing
  4. statement or the justification for not allowing the ethics committee. According to which regulations the tests were carried out.
  5. please check for typos, eg line 119: "... and 42n ..." should be 42nd. Superscript. Likewise, there are such errors throughout the text.
  6. why was the starting temperature so high? 35C? To my knowledge it should be 30-32C.
  7. line 131. starvation 12h - quite long, explain.
  8. The methodology is rather "sparse" described. I would recommend more detailed descriptions.
  9. in the methodology, please provide the formulas used to calculate adg, adfi etc.
  10. table 7. abbreviations should be described. Please provide the names of the fatty acids.
  11. Discussion - well done
  12. conclusions - please provide practical conclusions. How this research can influence the practical aspect of the activities.

The work is interesting, but I recommend these changes. I am not Native English Speaker, but I would recommend an English check. 

Sincerely

Reviewer 3 Report

My recommendation is Major revisions based on missing specific statements and conclusions as well as editing for the English language. However, this study does not appear to have significant lapses of scientific rigor or any other cause for rejection.

Reviewer 4 Report

Line 44: the reference seems unrelated to the statement. However, it seems more related to the next sentence.

Line 119: 42nd

Line 238: besides

Line 261: any justification how this BA redirect the nutrient to breast muscle growth instead of other tissues? And how fat type affect the breast muscle growth.

Line 275: Until

Round 2

Reviewer 3 Report

This novel report is now decently presented for publication. Moderate English language corrections would help the clarity of the manuscript.